# Effect of Total Starch and Resistant Starch in Commercial Extruded Dog Foods on Gastric Emptying in Siberian Huskies

**DOI:** 10.3390/ani11102928

**Published:** 2021-10-10

**Authors:** Taylor L. Richards, Alexandra Rankovic, John P. Cant, Anna K. Shoveller, Jennifer L. Adolphe, Dan Ramdath, Adronie Verbrugghe

**Affiliations:** 1Centre for Nutrition Modeling, Department of Animal Biosciences, University of Guelph, 50 Stone Road East, Guelph, ON N1G 2W1, Canada; tricha16@uoguelph.ca (T.L.R.); jcant@uoguelph.ca (J.P.C.); ashovell@uoguelph.ca (A.K.S.); 2Department of Clinical Studies, Ontario Veterinary College, University of Guelph, 50 Stone Road East, Guelph, ON N1G 2W1, Canada; arankovi@uoguelph.ca; 3Veterinary Biomedical Sciences, University of Saskatchewan, Saskatoon, SK S7N 5B4, Canada; Jennifer.Adolphe@adm.com; 4Guelph Research and Development Centre, Agriculture and Agri-Food Canada, 93 Stone Road West, Guelph, ON N1G 5C9, Canada; dramdath@uoguelph.ca

**Keywords:** canine, carbohydrates, grain-free, pet food, pharmacokinetic mathematical modelling, acetaminophen tracking

## Abstract

**Simple Summary:**

Gastric emptying is the release of nutrients from the stomach into the small intestine. The rate at which gastric emptying occurs may be associated with diabetes and obesity risk in humans and could help prevent weight gain in dogs. The largest portion of carbohydrates in pet diets is provided by various starches that are digested and absorbed at different rates. This study investigated the effects of common starch ingredients found in commercial dog foods on the gastric emptying rate in dogs. Dogs received each test diet once (4 total) and a glucose control twice in a randomized order, along with acetaminophen. Blood samples were taken once prior to meal consumption and at multiple time points after to determine acetaminophen concentrations. A mathematical model was used to estimate the rate of gastric emptying using postprandial acetaminophen concentrations. Overall, more gastric emptying occurred at a faster rate in dogs when fed the diet containing the highest fraction of starch ingredients resistant to canine digestion. These findings suggest that the inclusion of different starch sources may be associated with altered digestion and absorption of nutrients, which consequently affects gastric emptying rate. The link between carbohydrate sources and gastric emptying may provide a mechanism to prevent weight gain in dogs.

**Abstract:**

Gastric emptying rate (GER) may impact diabetes and obesity in humans and could provide a method to reduce canine weight gain. Starch, the most common source of carbohydrates (CHOs) in pet food, is classified as rapidly or slowly digestible, or resistant to digestion. This study investigated starch source effects in commercial extruded dog foods on the GER of 11 healthy adult Siberian Huskies. Test diets were classified as traditional, grain-free, whole-grain, and vegan. Dogs received each diet once, a glucose control twice, and acetaminophen (Ac) as a marker for GER in a randomized, partially replicated, 6 × 6 Latin square design. Pre- and post-prandial blood samples were collected at 16 timepoints from −15 to 480 min. Serum Ac concentrations were assessed via standard spectrophotometric assays and fitted with a mathematical model to estimate parameters of GER. Parameter values were subjected to ANOVA, with period and treatment as fixed effects and dog as a random effect. More total emptying (*p =* 0.074) occurred at a faster rate (*p =* 0.028) in dogs fed the grain-free diet, which contained the lowest total starch (34.03 ± 0.23%) and highest resistant starch (0.52 ± 0.007%). This research may benefit future diet formulations to reduce the prevalence of canine weight gain.

## 1. Introduction

Obesity is the most common nutrition-related disorder facing the domestic dog population today and can affect quality and longevity of life [1,2]. Obesity can be described as the excess accumulation of adipose tissue within the body and is usually the result of either excessive dietary intake or inadequate energy utilization, which leads to a state of positive energy balance [3]. Increased risk and aggravation of numerous co-morbidities, including cardiac, respiratory, and orthopedic diseases, cancer, and diabetes mellitus, have been associated with this disorder [4].

The rate of gastric emptying (GE) and the concentrations of postprandial glucose (PPG), which are directly associated in humans, may contribute to the onset of obesity and its secondary complications in canines [5,6]. Gastric emptying rate (GER) is defined as the speed with which nutrients are released from the stomach into the duodenum of the small intestine following ingestion [7]. A faster GER results in higher PPG concentrations [5]. If PPG concentrations exceed the capacity of insulin to clear glucose into the tissues through insulin dependent glucose transporters, this may lead to lipogenesis, which will ultimately contribute to weight gain and increased risk for obesity [5]. Although this relationship has predominantly been observed in human and rat models, a similar relationship is thought to exist in dogs [5,6] and was the impetus for the present study.

Carbohydrates (CHOs) are the most prevalent macro nutrient found in dry extruded pet foods (~40–60% dry matter) [8,9]. The largest proportion of CHOs in pet food is provided by starch [10], which is made up of two glucose polymers: amylose and amylopectin [11]. Amylose is less susceptible to digestion due to its linear structure while amylopectin has a branched chain structure, making it more susceptible to digestion [11]. The ratio of amylose to amylopectin varies across starch sources; most starch sources contain 20–35% amylose [12,13,14], while the amylose content of legume starches ranges from 29–65% [13,15]. Due to the differences in proportion of amylose, starch sources can be classified based on their rate of digestion: rapidly digestible; slowly digestible; or resistant to digestion [8]. Resistant starch (RS) refers to the portion of starch that cannot be enzymatically digested [16]. The glycemic index (GI) was developed as a way to rank CHOs based on their acute postprandial glycemic response in comparison to a control food (white bread or a standard glucose solution) [17]. More RS is hypothesized to result in lower PPG and therefore lower GI values due to the less digestible nature of this starch. In comparison, less RS means more digestible starch, which results in an increase in PPG and therefore GI. Additionally, RS enters the large intestine intact, where it is fermented by microorganisms [16]. This fermentation process is a source of volatile fatty acids (VFAs), which contribute to systemic energy metabolism in mammals [18] and also inhibit GE via the so-called ileo-colonic brake, a neuropeptidergic feedback mechanism controlling meal transit and digestion [19,20]. Due to the differences in proportion of these three starch fractions within foods and their subsequent variation in rate and degree of digestibility and fermentability, the effects of CHOs in pet food on GER are expected to vary.

In order to assess the rate of GE, several approaches have been investigated. Gamma Scintigraphy (GS) is considered by many to be the “gold standard” method to measure GE; however, this approach is fairly expensive and involves radiation exposure [7]. Acetaminophen (Ac) tracking is a method based on the understanding that Ac is slowly absorbed from the stomach but rapidly absorbed in the small intestine [21]. This approach is significantly less expensive than GS and involves no radiation exposure [7]. Additionally, a study completed by Glerup et al. in 2007 compared three different methods used to assess GE in research animals, including GS, Ac tracking, and the ^13^C-acetate breath test, and found that the results obtained from Ac tracking were similar to the results obtained via GS [22]. Consequently, Ac tracking is considered a reliable method to assess GER in mammals and was used in the present study [23].

Previous work by our team determined the PPG and GI values of the diets used in the present study and found that the order of GI from highest to lowest was: traditional grain diet > vegan diet > whole grain diet > grain-free diet [24]. Although numerical differences were observed, there were no significant differences in GI between dietary treatments [24]. The same study found that the grain-free diet induced the lowest area under the PPG curve during 30–480 min postprandial compared to the other dietary treatments, although these results were not significant [24]. Therefore, it was predicted that the grain-free diet, which results in the lowest PPG and GI values, should induce the slowest GER. Overall, the current literature discussed above suggests a link between the type of CHO consumed, its resulting GER, and subsequent GI and PPG. To the author’s knowledge, there is a dearth of research on this topic using a dog model. Determining the effects of CHOs in pet foods on GER and, in turn, PPG and GI values may be paramount in preventing weight gain and ultimately reducing the prevalence of conditions such as obesity in dogs. This study aimed to elucidate the relationship between starch sources found in various commercial extruded dog diets and the GER in healthy adult Siberian Huskies. Based on the aforementioned studies, authors hypothesized that the presence of digestible starch in the diet will increase PPG, GI, and GER, whereas RS will decrease PPG, GI, and GER.

## 2. Materials and Methods

### 2.1. Animals

The experimental protocol for this research was approved by the University of Guelph Animal Care Committee (AUP#3650) and in accordance with national and institutional guidelines for the care and use of animals. Eleven adult, client-owned Siberian Husky dogs (*n* = 4, neutered males; *n* = 5, spayed females; *n* = 2, intact females) were used in this study. Dogs were housed at an off-site facility (Rajenn Siberian Huskies, Ayr, ON, Canada) that had been visited and approved by the University of Guelph’s Animal Care Services. The dogs had a mean age of 5.63 ± 0.72 years (range 1.00–10.67 years) and a mean body weight (BW) of 23.32 ± 1.15 kg (range 19.00–30.68 kg). Body condition scores (BCS) for the dogs ranged between 3 and 6 (mean ± SD: 4.91 ± 0.63) on a 9-point-scale [25]. All dogs were deemed healthy based on medical history, physical examination, complete blood count (CBC), and serum biochemistry profile. Dogs were excluded if they had received medications 6 months prior to enrollment, had abnormalities on their physical examination, CBC, or serum biochemistry, or were younger than one year of age. The dogs were housed together in a group housing system and were separated and handled individually on study days. All dogs were transitioned onto the same diet (GO! FIT + FREE Adult Dog Food, Petcurean Pet Nutrition, Chilliwack, BC, Canada) 2 months prior to the start of the investigation. Dogs continued to eat this background diet throughout the entire study period. Dogs were fed the background diet in amounts suitable to maintain BW based on their dietary history. Body weight and BCS were recorded at each study visit and the amount fed was adjusted to maintain BW as required.

### 2.2. Dietary Treatments

Four commercially available dog diets were used as the test food in this study: Purina^®^ Dog Chow^®^ (Nestle Purina Petcare, St. Louis, MO, USA) as the traditional diet, GO! SENSITIVITY + SHINE™ Limited Ingredient Duck Recipe (Petcurean Pet Nutrition, Chilliwack, BC, Canada) as the grain-free diet, SUMMIT™ Three Meat Adult Recipe (Petcurean Pet Nutrition, Chilliwack, BC, Canada) as the whole-grain diet and Natural Balance^®^ Vegetarian Formula (Dick Van Patten’s Natural Balance Pet Foods, Burbank, CA, USA) as the vegan diet. Three diets were classified based on the main starch sources that were listed on their ingredient panels: traditional grain (corn, wheat), whole grains (oats, rye), grain-free (peas, lentils); the fourth diet vegan had no animal ingredients (Table 1). A 50% (wt/vol) glucose solution was used as the control.

The proximate analyses of the four commercial diets were performed by Central Testing Laboratories Ltd. (Winnipeg, MB, Canada) (Table 1). Proximate analyses of the diets were performed as follows, according to the Association of Official Analytical Chemists (AOAC) and the American Oil Chemist Society (AOCS) methods: ash by gravimetry (AOAC 923.03), crude protein by combustion (AOAC 990.03), crude fat by extraction (AOCS Am 5-04), crude fiber by gravimetry (AOCS Ba6a-05), total dietary fiber (TDF) by gravimetry (AOAC 991.43, 985.29), and moisture by gravimetry (AOAC 930.15).

Each dog received an amount of each dietary treatment that provided 25 g of available CHO (AvCHO) (62 g of the traditional diet, 77 g of the whole-grain diet, 65 g of the grain-free diet, and 55 g of the vegan diet), as determined by free sugar and starch analysis (Table 2). The quantity of AvCHO within the food was determined directly using the following equation (Food and Agriculture Organization of the United Nations 2003):*AvCHO = weight in grams [monosaccharides + disaccharides + oligosaccharides + [polysaccharides − fiber]].*

The total starch (TS) and RS content of each diet were determined enzymatically using commercially available assay kits (Megazyme International, Wicklow, Ireland; AOAC Method 996.11 and 2002.02, respectively) (Table 1). Free sugar content was determined via High Performance Liquid Chromatography (HPLC) as previously described by Brummer et al. (2015) (Table 1) [28]. The GI values of the traditional, whole grain, grain-free, and vegan diets, as reported previously by Rankovic et al. (2020), were 77 ± 18, 61 ± 7, 50 ± 13, and 71 ± 17, respectively.

### 2.3. Study Design 

This study was conducted according to a partially replicated 6 × 6 Latin square design where each dog received each commercial diet once and a 50% (wt/vol) glucose solution as a control that each dog received twice. Between trials, there was a 7-day washout period during which only the background diet was fed.

Dogs were fasted overnight (14 h) prior to each test. On test days, dogs were weighed and BCS was recorded. EMLA cream (2.5% Lidocaine) was applied to the dogs’ legs at the site where a 20 Ga IV catheter (Insyte-W 20GA × 1.1, Becton Dickinson, Franklin Lakes, NJ, USA) was placed into a cephalic or saphenous vein. Once placed, catheters were flushed immediately with 2 mL of 0.9% sodium chloride solution (Baxter International, Deerfield, IL, USA), followed by 0.1 mL of 4.0% sodium citrate (Baxter International, Deerfield, IL, USA) to prevent clotting. Prior to blood collection, catheters were flushed with 0.5 mL sterile isotonic sodium chloride solution, and 0.5 mL of blood was withdrawn and discarded to avoid dilution. Dogs were allowed a minimum of 15 min to recover from catherization. The first blood sample occurred at −15 min (baseline) prior to feeding the test diet or glucose control with the oral dose of Ac (16.30 mg/kg BW) administered immediately following feeding. Timing began as soon as the dog began ingesting the test diet or glucose control and blood samples were taken at 15, 30, 45, 60, 90, 120, 150, 180, 210, 240, 270, 300, 360, 420, and 480 min postprandial. At each time point 2.5 mL of blood was collected into serum separation tubes (Vacutainer™, Becton Dickinson, Franklin Lakes, NJ, USA) and immediately refrigerated (4 °C). After collection, catheters were flushed with 2 mL of sodium chloride and 0.1 mL of sodium citrate to maintain patency. The samples were centrifuged at 4 °C at 1000× *g* for 10 min (Legend RT, Kendro Laboratory Products, Asheville, NC, USA), aliquoted and serum was stored in a freezer at −20 °C until analysis.

### 2.4. Acetaminophen Assay and Gastric Emptying Model

Serum Ac content was determined using a commercially available spectrophotometric assay kit (Paracetamol Assay Kit K8002, Cambridge Life Sciences, Cambridgeshire, UK). The assay was performed in triplicate according to the manufacturer’s instructions and run on a Powerwave Xs Microplate Spectrophotometer (Biotek, Winooski, VT, USA) with KC4 Data Analysis Software (Biotek, Winooski, VT, USA) (Wavelength = 650 nm).

The mathematical model used to estimate GE parameters from the serum Ac time course has been described previously by Stahel et al. (2016) [21]. Briefly, Ac is assumed to enter serum by first-order GE according to rate constant k_SB_, and exit serum according to rate constant k_el_, to yield the following differential equation:*dAc_Se_/dt = k_SB_ × Ac_St_ − k_el_ × Ac_Se_*
where Ac_St_ and Ac_Se_ are the masses of Ac (mg) in stomach and serum, respectively. For each sampling interval, k_SB_ is assigned a value of 0, k_SB2_, or k_SB3_, representing no, slow, or fast emptying, respectively, based on the observed rate of change in serum Ac concentration (ΔcAc_Se_). Accordingly, k_SB_ = 0 when ΔcAc_Se_ < −0.05 mg/L/min; k_SB_ = k_SB2_ when −0.05 mg/L/min ≤ ΔcAc_Se_ ≤ 0.05 mg/L/min; and k_SB_ = k_SB3_ when ΔcAc_Se_ > 0.05. The predicted concentration of Ac in serum (cAc_Se_) was calculated from Ac_Se_ assuming a volume of distribution of 50% of BW. Best-fit values of k_SB2_, k_SB3_, and k_el_ were estimated with the Solver function of Microsoft Office Excel 2020 Version 16.37 to minimize residual sums of squares between predicted and observed cAc_Se_ above baseline. Goodness of fit for each curve is presented as the root mean square prediction error (rMSPE) as a percentage of the mean observed:$$\mathrm{rMSPE}\%=\frac{\sqrt{\sum_{i=1}^{n}{\left(predicted_{i}-observed_{i}\right)}^{2}/n}}{\sum_{i=1}^{n}{\mathrm{observed}}_{i}/n}$$
where *n* is the number of observations in the time course. Figure 1 provides an example of predicted and observed ΔcAc_Se_ curves used in this study for one dog on one treatment with a typical rMSPE of 16.76% of the mean. Sums were calculated of the amount of time spent emptying rapidly (time-fast) and slowly (time-slow) as well as the amount of time with no emptying (time-off). A total emptying index to amalgamate time and rate of GE was calculated as k_SB2_ × time-slow + k_SB3_ × time-fast.

### 2.5. Statistical Analysis

Curve fits with rMSPE values greater than 50% were deemed unreliable and not included in further analysis (13 total: glucose control, 6; traditional, 1; grain-free, 3; whole-grain, 1; and vegan, 2). Parameters of the remaining fitted curves were subjected to analysis of variance (ANOVA) using the MIXED procedure of SAS (Statistical Analysis System, Version 9.4, Cary, NC, USA), where period and treatment were fixed effects and dog was random. Results are presented as means with a pooled SEM over all treatments. When *p* ≤ 0.10 for the overall treatment effect, significance of orthogonal contrasts between treatment means was declared at *p* ≤ 0.05, and trends were declared at 0.05 < *p* ≤ 0.10.

## 3. Results

All statistical analyses values discussed below are displayed in Table 3.

### 3.1. First-Order Slow Gastric Emptying Rate Constant (k_SB2_)

A trend was observed in the effect of treatment on k_SB2_ (*p* = 0.077). Following ingestion of the grain-free diet, k_SB2_ was greater compared to the traditional diet (*p* = 0.021) and the glucose control (*p* = 0.028).

### 3.2. First-Order Fast Gastric Emptying Rate Constant (k_SB3_)

An effect of treatment (*p* = 0.028) was noted on k_SB3_, with the glucose control inducing lower k_SB3_ values compared to both the whole-grain (*p* = 0.018) and grain-free diets (*p* = 0.011).

### 3.3. First-Order Elimination Constant (k_el_)

There was no effect of treatment on k_el_ (*p* = 0.386).

### 3.4. Area under the Serum Acetaminophen Curve (AUC)

The area under the serum Ac curve (AUC) was affected by treatment (*p* < 0.001). The AUC was greater following consumption of the traditional (*p* = 0.003), whole-grain (*p* < 0.001), grain-free (*p* < 0.001), and vegan (*p* < 0.001) diets compared to the glucose control. Additionally, the vegan diet induced a higher AUC in contrast to the traditional diet (*p* = 0.043). Furthermore, a trend was observed in that ingestion of the grain-free diet resulted in a higher AUC compared to the traditional diet (*p* = 0.067).

### 3.5. Total Emptying Index

A trend was also observed in the effect of treatment on total emptying (*p* = 0.074). Total emptying was greater in dogs following consumption of the grain-free diet compared to traditional diet (*p* = 0.045) and glucose control (*p* = 0.023).

### 3.6. Time-Off, Time-Slow, and Time-Fast

No treatment effect was found for time-off, time-slow, or time-fast values (*p* = 0.278, *p* = 0.726, and *p* = 0.077 respectively).

## 4. Discussion

The results presented herein suggest that the grain-free diet, which included peas and lentils as the primary starch sources, induced both the greatest amount of GE and the most rapid GER. These findings may be attributed to the proportional differences between slowly digestible, rapidly digestible, and RS in the test diets.

In human trials, pulse ingredients, such as lentils, peas, and chickpeas, significantly attenuate PPG and are consequently categorized as low GI foods [29]. Pulse ingredients contain higher amounts of slowly digestible starch (SDS) and RS, and lower amounts of rapidly digestible starch (RDS) compared to cereal grains, which contain lower concentrations of SDS and RS and higher concentrations of RDS [29]. The relative proportions of SDS, RDS, and RS in pulse ingredients have been attributed to their high amylose content compared to cereal starches, strong interactions between amylose chains, the presence of intact cell structures after cooking, high dietary fiber components, and anti-nutrients [29]. The differences in starch profiles are thought to subsequently result in lower rates of hydrolysis, PPG, and GI following ingestion of pulses compared to cereal grains [29]. The proportions of SDS, RDS, and RS may influence the GER of a diet as well, as SDS slows GE [30]. For example, as mentioned previously, when RS enters the large intestine, it is fermented, resulting in the production of VFAs, which have been shown to inhibit GE via the ileo-colonic brake [18,19]. Taken together, diets with more pulse ingredients, thereby containing the greatest proportion of SDS and RS, and lowest proportion of RDS, should result in the lowest PPG, GI, and GER.

In the present study, the grain-free diet contained the highest amount of RS and lowest amount of TS relative to the other diets. Furthermore, the grain-free diet contained the most pulse ingredients, indicating that it also contained the highest amount of SDS and least amount of RDS. In comparison, the vegan, traditional, and whole-grain diets had less RS, more TS, and less pulses ingredients, indicating less SDS and more RDS. Based on these findings, it was hypothesized that the grain-free diet would result in the lowest PPG, GI, and GER due to the higher amount of pulse ingredients. Rankovic et al. (2020) indeed found that the grain-free diet used in the present study had the lowest GI value compared to the traditional, whole-grain, and vegan diets, although these results were not significantly different [24]. The same study found that the grain-free diet induced the lowest area under the PPG curve between 30–480 min compared to the other dietary treatments [24]. The lower PPG concentration induced by the grain-free diet was only significant when compared to the whole grain diet and the glucose control between 0–150 min [24]. However, contrary to the authors hypothesis for the present study, the grain-free diet, which contained the highest concentration of RS and induced the lowest GI and PPG values, resulted in the fastest GER.

One possibility for these contradictory results in dogs may be due to the effect that RS has on the osmoreceptors found in the duodenum [31,32]. Gastric emptying is thought to be slowed when these receptors are stimulated by material arriving from the stomach, specifically by the products of starch (sugars) and protein (amino acids) breakdown, along with the soaps formed during fat digestion [31,32]. As mentioned previously, RS is not digested in the gastrointestinal tract; instead, it passes relatively intact through the stomach, small intestine, and colon [29]. Therefore, it is possible that the elevated amount of RS in the grain-free diet may have resulted in fewer starch breakdown products being emptied into the small intestine. The duodenal receptors slow GER when triggered. However, if the grain-free diet had fewer breakdown products, then this process does not occur as frequently compared to the other diets. In comparison, the diets containing less RS may have induced a larger amount of starch breakdown products to enter the intestine, stimulating the receptors to slow GE.

A second possible explanation for the seemingly contradictory results may be related to the high protein/high fat background diet that the dogs received during the acclimation period two months prior to the start of the study and on days between treatments. Duodenal receptors are thought to slow GE when triggered by the breakdown of fat, protein, and starch. However, the 25 g AvCHO portion of the grain-free diet, which contained the greatest amount of fat and protein compared to the vegan and traditional diet portions, induced the fastest GER. Research completed in rats and humans suggests that the consumption of a high protein/high fat diet for as little as 2 weeks accelerates the GER of a different high protein/high fat test diet significantly compared to a low protein/low fat test diet [33,34]. Additionally, French et al. (2007) found that Cholecystokinin (CCK), a potent inhibitor of GE, usually stimulated by protein and fat entering the intestine, increased following the ingestion of a high fat diet for 2 weeks [35]. However, this did not result in a reduction in GER [35]. Together, the data from these studies suggest that subjects may be desensitized to the effects of CCK following a period of high protein/high fat consumption. It is possible that after consuming the high protein/high fat background diet for two months prior to the start of the present study, the dogs became desensitized to the effects of CCK, allowing the higher protein and fat grain-free diet to accelerate GER. Although the 25 g AvCHO portion of the grain-free diet provided slightly less protein and fat than the whole-grain diet which induced slower emptying, the difference in protein and fat between diets was negligible. Additionally, the slightly slower emptying for the whole-grain diet may also be attributed to the higher RDS and lower RS and SDS as discussed above, considering the difference in starch composition was greater than the difference in fat and protein content between the whole-grain and grain-free diets. This may lend some indication as to why the k_SB3_ value for the grain-free diet was significantly greater than the k_SB3_ value for the vegan and traditional diets, which had the lowest amounts of protein and fat per 25 g AvCHO portion compared to the grain-free diet. This hypothesis cannot be substantiated with certainty, as the starch composition of the background diet was not analyzed, but does warrant future investigations to study the effect of dietary history and macro nutrient composition on GER in dogs.

The varying levels of fiber in the test diets could have also affected the GER. Fiber naturally present in food can delay the GER of a solid meal [36]. Soluble fiber has a much higher water holding capacity than insoluble dietary fiber, causing soluble-fiber rich foods to increase viscosity in the stomach, delaying emptying into the small intestine [37]. Soluble fiber is also fermented, resulting in production of VFAs, which in turn inhibit GE indirectly via the ileo-colonic brake [18,19]. Peptide YY, also known as peptide tyrosine-tyrosine (PYY) has been suggested to be driving this humoral pathway [18,19]. When comparing all treatments, the 25 g AvCHO portion of the grain-free diet, which induced the highest GER, contained the highest amount of TDF compared to the vegan and traditional diet portions, which had the lowest amount of TDF. Additionally, the 25 g AvCHO portion of the grain-free diet had the greatest amount of crude fiber compared to the traditional, vegan, and whole grain diet portions. Total dietary fiber includes both soluble and insoluble fiber fractions, while crude fiber consists only of insoluble fiber [38]. The difference in crude fiber was larger than the difference in TDF, suggesting higher insoluble fiber in the grain-free diet, which could have resulted in less viscosity in the stomach and lower VFA production in the distal intestine, and thus faster GER compared to the other treatments. In contrast, the larger amount of soluble fiber in the other diets could result in more viscosity in the stomach and augment VFA production through fermentation, consequently slowing GE into the intestine; this has been previously demonstrated in human studies [18,19,37,38]. Less soluble fiber could have contributed to the higher k_SB3_ value observed for the grain-free diet. The grain-free diet had less TDF compared to the whole-grain diet; however, similar to the fat and protein content of these diets, this difference is negligible and the difference in starch profiles between the diets likely played a larger role in the observed results. This hypothesis cannot be substantiated as the test diets were not specifically analyzed for soluble and insoluble fiber fractions.

Nutrients entering the duodenum following meal ingestion activate a proximal-distal neuroendocrine loop which stimulates glucagon-like peptide-1 (GLP-1) secretion [39]. Glucagon-like peptide-1 stimulates insulin secretion and inhibits glucagon, which is responsible for raising concentrations of glucose in the blood stream [39]. As discussed above, the 25 g AvCHO portion of the grain-free diet had greater protein and fat being broken down and entering the duodenum compared to the traditional and vegan diets, which could stimulate more GLP-1 production and therefore less glucagon, contributing to a lower PPG. This could explain why the diet with the fastest GER induced the lowest PPG. More research is warranted to better understand the relationship between GER, PPG, and GLP-1 in dogs. Additionally, it may be of interest to investigate the effects that this correlation has on satiety and appetite control, as GLP-1 promotes satiety and can reduce food intake in humans [39]. This could be an important contributing factor to weight gain/obesity risk.

A limitation of this study is that the dogs only consumed a portion of each diet that provided 25 g of AvCHOs. This was necessary in order to concomitantly perform GI testing. However, future research is necessary to investigate the effect of consumption of a full meal, as this would alter the amount of nutrient digestion and absorption in the gastrointestinal tract, and could subsequently affect GER.

While the data presented herein helps to improve our understanding of the effects of different diets on GER in dogs, future research investigating the effect of macronutrients on GE using a dog model is warranted. Canine GE should continue to be investigated in a variety of breeds of dog, including breeds larger than research beagles, such as the Siberian Huskies used in the current study. This is in order to establish correct dog size and breed-specific observations related to GE. Gastric emptying rate is affected by breed and size, in that larger dog breeds have a longer GE time compared to smaller breeds, possibly due to the difference in size/length of their gastrointestinal tract [7]. Siberian Huskies, the breed used in the current study, have reduced copy numbers of the gene coding for pancreatic Alpha-amylase 2B (AMY2B), which correlates with serum amylase activity in dogs [40,41]. In contrast, most variation in AMY2B copy numbers was noted in Beagles, the breed most often used for research [40,41]. It is unclear, however, how this impacts CHO digestion and absorption and how this could impact GER. Moreover, glucose transport activity may be different in Siberian Huskies competing in long-distance races. The Siberian Huskies participating in the current study did not compete, nor were they trained for such high intensity activities. The dogs did not participate in races or training during the research and at least 2 months prior. The small sample size of 11 dogs is an additional limitation within this study. Although this is partially remedied by the crossover design used, authors recommend future research be carried out using a larger sample size.

## 5. Conclusions

The results of this study indicate that the grain-free diet, which had the highest amount of RS and SDS and lowest amount of TS and RDS, induced more overall GE at a more rapid rate in Siberian Huskies compared to the traditional, vegan, and whole-grain diets, and the glucose control. Authors hypothesize that these findings may be due to the differences in macro nutrient content of the test foods, including the composition of the starch sources. The fat and protein content of the background diet and test diets, as well as the fiber content of the test diets may have also contributed to the observed results. It may be of interest to further explore the relationship between GER and PPG, specifically in regard to CCK, GLP-1, and PYY, which play a role in satiety and appetite control. Additionally, the humoral pathway linking GER and microbial fermentation of CHO and fiber sources in the distal intestine needs further investigation. This research may be paramount in determining how to prevent weight gain, and ultimately reduce the prevalence of obesity in dogs.

## Figures and Tables

**Figure 1 animals-11-02928-f001:**
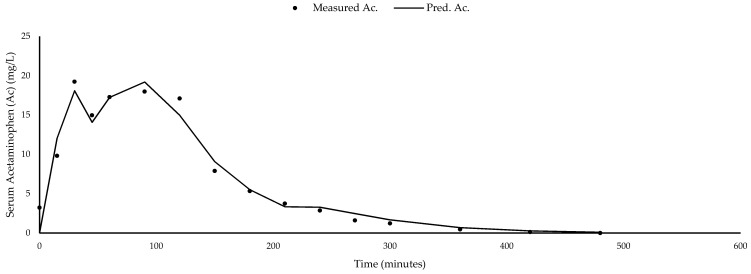
Predicted (solid line) and observed (circles) serum acetaminophen concentration curves in a gastric emptying study using acetaminophen as a tracer in dogs over a 480-min period. Predicted serum acetaminophen concentrations were determined using a mathematical model, described previously by Stahel et al. [21]. rMSPE% of curve = 16.76%.

**Table 1 animals-11-02928-t001:** Proximate analysis, total dietary fiber, and energy contents, as well as in vitro starch and free sugar contents of four commercial extruded dog foods containing different starch sources fed in a gastric emptying trial to 11 client-owned Siberian Huskies.

	Treatment			
	Traditional	Whole Grain	Grain-Free	Vegan
Moisture (%)	5.85	6.32	6.56	7.15
Crude Protein (%DM)	25.55	26.10	28.59	23.46
Crude Fat (%DM)	10.25	10.21	11.58	8.20
Ash (%DM)	7.24	10.10	7.66	5.00
Crude Fiber (%DM)	1.05	1.24	3.14	2.10
NFE ^1^ (%DM)	55.91	52.35	49.03	61.24
TDF (%DM)	8.10	8.80	10.3	10.0
GE (kcal/kgDM) ^2^	4712	4593	4728	4619
ME (kcal/kgDM) ^3^	3959	3841	3816	3846
AvCHO (%DM)	42.51	41.13	34.63	48.52
Total Starch (%DM)	41.69 ± 1.48	40.83 ± 1.90	33.91 ± 0.44	48.06 ± 4.26
Resistant Starch (%DM)	0.41 ± 0.02	0.16 ± 0.01	0.56 ± 0.02	0.26 ± 0.03
Glucose (%DM)	0.099 ± 0.0032	0.023 ± 0.0008	0.032 ± 0.0013	0.029 ± 0.0018
Sucrose (%DM)	1.40 ± 0.027	0.56 ± 0.028	1.38 ± 0.022	0.87 ± 0.036

DM, dry matter; NFE, nitrogen free extract; TDF, total dietary fiber; GE, gross energy; ME, metabolizable energy; and AvCHO, available carbohydrate. Values reported on dry matter basis, except for moisture. ^1^ NFE = 100 − (weight in grams (protein + fat + ash + crude fiber) in 100 g of food). ^2^ GE (kcal/kgDM) = (5.7 × g protein) + (9.4 × g fat) + [4.1 × (g NFE + g crude fiber)] [26]. ^3^ ME (kcal/kgDM) = 575 + (0.816 × GE (kcal/kg)) + (12.08 × % fat) − (52.76 × % crude fiber) – (20.61 × % protein) – (6.07 × % moisture) [27]. Ingredient panels, starch sources in *italic*: **Traditional:**
*Whole grain corn*, meat and bone meal, corn gluten meal, beef fat preserved with mixed-tocopherols, soybean meal, poultry by-product meal, chicken, egg and chicken flavour, *whole grain wheat*, pork & poultry digest, salt, calcium carbonate, potassium chloride, mono and dicalcium phosphate, choline chloride. Vitamins [vitamin E supplement, niacin (vitamin B3), vitamin A supplement, calcium pantothenate (vitamin B5), pyridoxine hydrochloride (vitamin B6), vitamin B12 supplement, thiamine mononitrate (vitamin B1), vitamin D3 supplement, riboflavin supplement (vitamin B2), menadione sodium bisulfite complex (vitamin K), folic acid (vitamin B9), biotin (vitamin B7)] Minerals [zinc sulphate, ferrous sulphate, manganese sulphate, copper sulphate, calcium iodate, sodium selenite] l-lysine monohydrochloride yellow 6, yellow 5, red 40, soybean oil, blue 2, garlic oil. **Whole grain:** Chicken meal, *oatmeal*, *whole brown rice*, *rye*, *barley*, chicken fat (preserved with mixed tocopherols), salmon meal, lamb meal, natural chicken flavour, whole dried egg, rice bran, dried kelp, flaxseed, dicalcium phosphate, calcium carbonate, potassium chloride, choline chloride, l-lysine, sodium chloride. Vitamins [vitamin A supplement, vitamin D3 supplement, vitamin E supplement, niacin, l-ascorbyl-2-polyphosphate (a source of vitamin C), d-calcium pantothenate, thiamine mononitrate, beta-carotene, riboflavin, pyridoxine hydrochloride, folic acid, biotin, vitamin B12 supplement] Minerals [zinc proteinate, iron proteinate, copper proteinate, zinc oxide, manganese proteinate, copper sulphate, ferrous sulphate, calcium iodate, manganous oxide, selenium yeast] taurine, dl-methionine, dried rosemary. **Grain-free:** De-boned duck, duck meal, *peas*, *lentils*, *tapioca*, *pea flour*, canola oil (preserved with mixed tocopherols), *chickpeas*, natural flavour, coconut oil (preserved with mixed tocopherols), monocalcium phosphate, salmon oil, sodium chloride, calcium carbonate, potassium chloride, dried chicory root, choline chloride. Vitamins [vitamin A supplement, vitamin D3 supplement, vitamin E supplement, niacin, l-ascorbyl-2-polyphosphate (a source of vitamin C), d-calcium pantothenate, thiamine mononitrate, beta-carotene, riboflavin, pyridoxine hydrochloride, folic acid, biotin, vitamin B12 supplement] Minerals [zinc proteinate, iron proteinate, copper proteinate, zinc oxide, manganese proteinate, copper sulphate, ferrous sulphate, calcium iodate, manganous oxide, selenium yeast] taurine, dried rosemary. **Vegan:**
*Brown rice*, *oat groats*, *barley*, *peas*, potato protein, canola oil (preserved with mixed tocopherols), *potatoes*, dicalcium phosphate, dried tomato pomace, natural flavor, calcium carbonate, potassium chloride, choline chloride, taurine. Vitamins [vitamin E supplement, vitamin A supplement, d-calcium pantothenate, niacin, riboflavin supplement, vitamin D2 supplement, vitamin B12 supplement, thiamine mononitrate, pyridoxine hydrochloride, folic acid, biotin] Minerals [zinc proteinate, zinc sulfate, ferrous sulfate, iron proteinate, copper sulfate, copper proteinate, manganese sulfate, manganese proteinate, calcium iodate, sodium selenite] salt, flaxseed, dried spinach, parsley, cranberries, l-lysine monohydrochloride, l-carnitine, citric acid (used as a preservative), mixed tocopherols (used as a preservative), yucca schidigera extract, dried kelp, l-ascorbyl-2-polyphosphate (source of vitamin c), rosemary extract.

**Table 2 animals-11-02928-t002:** Macro nutrient content of four commercial extruded dog foods containing different starch sources fed in a meal response test to 11 client-owned Siberian huskies, expressed as the dry-matter quantity in grams that each dog received to provide 25 g available carbohydrate of each diet.

	Treatment			
	Traditional	Whole Grain	Grain-Free	Vegan
Portion size for 25g AvCHO (g)	62	77	65	55
Crude Protein (g)	15.84	20.09	18.58	15.46
Crude Fat (g)	6.35	7.86	7.52	5.41
Crude Fiber (g)	0.651	0.954	2.04	1.15
TDF (g)	5.02	6.77	6.69	5.50

AvCHO, available carbohydrate; TDF, total dietary fiber; Values reported on dry matter basis.

**Table 3 animals-11-02928-t003:** Markers of gastric emptying, via acetaminophen tracking, in a meal response test in 11 client-owned Siberian Huskies fed four commercial extruded dog foods containing different starch sources in amounts to provide 25 g available carbohydrate of each diet.

	Treatment						
	Traditional	Whole Grain	Grain-Free	Vegan	Glucose	SEM	*p*-Value
k_SB2_, /min	0.0010 ^a^	0.0075 ^ab^	0.0164 ^b^	0.0123 ^ab^	0.0038 ^a^	0.0031	0.077
k_SB3_, /min	0.0137 ^ab^	0.0251 ^a^	0.0270 ^a^	0.0184 ^ab^	0.0115 ^b^	0.0035	0.028
k_el_, /min	0.0154	0.0133	0.0114	0.0105	0.0156	0.0020	0.386
rMSPE, %	28.3	28.9	29.4	32.8	34.8	2.7	0.319
Time-off, min	125	142	154	153	109	15	0.278
Time-slow, min	251	258	266	233	286	23	0.726
Time-fast, min	105	80	59	94	85	12	0.774
AUC, mg·min/L	2369 ^b^	2638 ^bc^	2896 ^bc^	2879 ^c^	1653 ^a^	228	<0.001
Total emptying index	1.88 ^b^	4.05 ^ab^	5.77 ^a^	4.78 ^ab^	1.94 ^b^	0.93	0.074

k_SB2_, first-order slow gastric emptying rate constant; k_SB3_, first-order fast gastric emptying constant; k_el_, first-order elimination constant; rMSPE, root mean square prediction error as percentage of observed mean; and AUC, area under the serum acetaminophen curve. Values expressed as mean ± SEM, standard error margin; *n* for each dietary treatment = 53 total: traditional, 10; whole-grain, 10; grain-free, 8; vegan, 9; glucose control, 16; ^a,b,c^ Values in a row with superscripts without a common letter differ significantly (*p* ≤ 0.05), repeated measures ANOVA. Treatment effects were estimated as preplanned orthogonal contrasts between all four treatments along with the glucose control. Results are presented as means with a pooled SEM over all treatments. Dogs received each commercial diet (traditional, grain-free, whole-grain, and vegan diet) once and a 50% (wt/vol) glucose solution as a control twice. Curves with rMSPE% values greater than 50% were deemed unfit and removed from analysis (13 total: glucose control, 6; traditional, 1; grain-free, 3; whole-grain, 1; and vegan, 2). Curves with rMSPE% values less than 50% were analyzed (52 total: glucose control, 16; traditional, 9; grain-free, 8; whole-grain, 10; and vegan, 9).

## Data Availability

The data presented in this study are available on request from the corresponding author.

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
