# Peer review of "Effect of Total Starch and Resistant Starch in Commercial Extruded Dog Foods on Gastric Emptying in Siberian Huskies"

_animals, 2021, doi:10.3390/ani11102928_

Round 1

Reviewer 1 Report

Limitations:
1. Small sample size, (11 dogs), which is partially remedied by the crossover design of the study to render a more robust data set.
2. Given the recent concerns about grain-free diets and possible association with cardiac disease, it seem remiss for the authors to not at least mention this in their introduction or discussion.  While I do understand that the objective of this study was to specifically investigate the effects of different commercial diets on gastric emptying, and more specifically, the role of starch in this process, the fact that the authors conclude that the grain-free diet may be beneficial to health (i.e. weight loss and prevention of obesity), should include a more in-depth and balanced discussion of the potential pros and cons of grain-free diets, including potential link to cardiac disease. 

Lines 305-306, and lines 308-309 : First order gastric emptying constant and first order elimination constant:  "A trend was observed in the effect of treatment...    What were these trends and how did it effect of treatment?

Strengths:
1. The manuscript is well-written and referenced, with the exception of discussion of adverse health effects of grain-free diets.
2. Use of the acetaminophen clearance test method vs. scintigraphy was an interesting methodology, and I appreciated the authors detailed explanation of why they chose this method and how if compares with the gold standard method of scintigraphy, as it offers a possibly more practical methodology for researchers for whom advanced technology for further research is may be limited.

Author Response

Dear Reviewer,

Thank you for taking the time to read this manuscript and provide valuable feedback. See attached for our responses to your comments. 

Thank you!

Reviewer 2 Report

Dear authors,

I read your work carefully and launch some reflections.

Title

The study has nothing to do with the function of dogs, so despite the team already having a work that refers to non-racing sled dogs, this designation makes no sense to me.

I suggest,...

Effect of total starch and resistant starch in commercial extruded dog
foods on gastric emptying in Husky Siberian dogs

or 

Effect of total starch and resistant starch in commercial extruded dog
foods on gastric emptying in a dog model

Introduction

Page 2, Line 78 - Fermentation in the intestine is a source of volatile fatty acids that play a number of important functions in the intestine that could be described.

Page 3, Line 116 - In a previous work by our team instead Rankovic et al. (2020)

At the end of the introduction, the objective of the work should be more clearly explained.

Materials and Methods

In my opinion the sample is very heterogeneous in age, weight, body condition. Why didn't they do this work with laboratory animals - Beagle dogs of the same age, sex, weight, body condition?

Page 4 and 5, Lines 184 to 187 and so on ...- Vitamin designations in capital letters

Vitamin is an essential substance for life that the body cannot synthesize, while the dog's body can synthesize ascorbic acid, so I am afraid that the designation of Vitamin C in companion animal medicine is not correct.

It is advisable to build a list of acronyms and abbreviations namely CCK.

p values should be written in italic

Author Response

(The authors gave the same response as above.)
